# Cybersex and Attachment Styles: Proposal of the Emotional and Relational Aspects in Cybersex Activities (ERACA) Questionnaire

**DOI:** 10.3390/ijerph20247151

**Published:** 2023-12-07

**Authors:** Andrea Baroncelli, Maria Giulia Taddei, Roberta Giommi, Elena Lenzi, Carolina Facci, Enrica Ciucci

**Affiliations:** 1Department of Philosophy, Social Sciences and Education, University of Perugia, 06123 Perugia, Italy; andrea.baroncelli@unipg.it; 2Independent Researcher, 51100 Pistoia, Italy; taddeimariagiulia@gmail.com; 3Istituto Internazionale di Sessuologia, Istituto Ricerca e Formazione, 50123 Florence, Italy; giommi.roberta@libero.it (R.G.); lenzielena64@gmail.com (E.L.); 4Department of Education, Languages, Intercultures, Literatures, and Psychology, University of Florence, 50121 Florence, Italy; enrica.ciucci@unifi.it

**Keywords:** cybersex, attachment, internet, quality of relationships, assessment

## Abstract

The current study presents the development and the initial validation of a new questionnaire to assess individual differences in emotional and relational aspects related to cybersex activities (i.e., the ERACA). A total of 246 adults (105 females, mean age = 31.89 years, SD = 10.03) coming from the general adult population participated in the study. The items of the ERACA were developed considering the extant literature, and an exploratory factor analysis approach indicated a three-factor structure (i.e., the gratification of the Self through the objectification of other people, the gratification of the Self through relational aspects, betrayal, and infidelity). The associations between the dimensions of the ERACA and dimensional measures of both attachment styles and online sexual behaviors indicated that different aspects related to the quality of the relationships play a different role in individual differences concerning emotional and relational aspects of cybersex activities. The discussion emphasizes the potential usefulness of the ERACA questionnaire for both research purposes and from a health-promoting point of view.

## 1. Introduction

The vast majority of the extant studies concerning the assessment of cybersex activities have focused on measuring time spent online and describing activities, revealing the motives that drive people in its use, or highlighting its potential pathological implications [1,2,3,4,5,6,7], while related emotional and relational aspects have been explored less extensively. Moving within a developmental psychology framework interested in highlighting these aspects within a community sample of adults, the aim of the present study was to provide the development and the initial validation of a new questionnaire specifically devoted to assessing individual differences in emotional and relational aspects related to cybersex activities; further, we paid specific attention to investigate the associations between the dimensions of the questionnaire (i.e., named Emotional and Relational Aspects in Cybersex Activities—ERACA) and dimensional measures of both attachment styles and online sexual behaviors.

In the last three decades, the Internet has rapidly become a popular accessible medium all over the world, providing quick exchange of a wide range of contents and information [8], and its use for sexual purposes has become extremely high [9]. In this regard, the advent of cyberspace has affected the trajectories through which people build relationships and develop their inner psychosocial processes. Considering the specific of human sexuality, over twenty years ago it was stated that the spread of the Internet was the cause of a new “sexual revolution” with positive effects on human well-being (e.g., the opportunity to meet and support members of sexual minority groups, the rapid spread of sexual education and information about safe sexual practices, etc.) coexisting with negative ones (e.g., online sexual compulsivity and online sexual addiction) [10]. In terms of psychological constructs, extant research defines online sexual activities (i.e., OSAs) as any arousal or non-arousal activity on the Internet that involves sexuality (e.g., text, audio, video, graphics, educational content), while the specific term “cybersex” identifies a subcategory of OSAs in which the Internet is used for achieving sexual arousal and gratification [1,2]. The engagement in OSAs occurs across the lifespan [1,6,9]; typically, the first episodes happen during preadolescence for curiosity about how sexual intercourse occurs, and pleasure seeking becomes relevant during adolescence and adulthood [11]. OSAs are not deterministically associated with maladaptive outcomes per se [12,13,14]. Nevertheless, the vast majority of the assessment practices related to OSAs and cybersex have considered their potential pathological aspects (e.g., the amount of time spent in the activity, the consequences in various areas of one’s own offline life, etc. [4,5,7]). In the present study, we focused on the emotional and relational aspects that are involved in cybersex, and we assessed individual differences in beliefs and behaviors concerning the following: (a) emotional activations related to cybersex (i.e., feelings of pleasure and satisfaction, or unpleasant emotional states); (b) the establishment of close contact or the maintenance of a relational distance with reference to the other involved people—whether they are the protagonists of materials used in solitary activities or the partners during synchronous interactions; and (c) the impact of cybersex on offline romantic relationships (i.e., to which extent cybersex is perceived as a form of betrayal and infidelity). Aware of the close continuity between emotional and relational aspects, we decided to focus on these three facets because they allow us to point to the whole cybersex experience without losing sight of (a) the internal emotional activations, (b) the role and the valence attributed to the other involved people, and (c) the specific beliefs about betrayal and infidelity.

First of all, we considered the different reasons that lead individuals to seek arousal by practicing cybersex. According to the classic classification by Cooper and colleagues [2], some people consider cybersex a simple means to reach distraction and gratification (i.e., the “recreational” users), while others are attracted by the perception of practicing forbidden sexual activities (i.e., the “fantasy-type” users) or to easily have success after previous frustrating sexual experiences (i.e., the “past difficulties with sex” users), and some others use cybersex as a way to regulate overwhelming unpleasant emotions and dysphoria (the “depressive-type” users) or to control stress (i.e., “stress-reactive users”). It therefore appears that the same cybersex activity can be related to different intra- and interpersonal processes for different individuals who practice it, thus making it essential to evaluate these differences through a specific tool. For instance, in assessing the emotional facets of cybersex activities people are involved in, it is central to understand to which extent they perceive rewarding and fulfilling arousal activation, as well as to which extent they experience unpleasant emotional states; further, it is important to investigate whether cybersex users perceive to use such practices as a mean to deal with their unpleasant moods and/or to seek pleasant ones. Moreover, specific attention must be paid to individual differences in the relational valence of cybersex. To do so, we referred to the growing body of research focused on OSAs and attachment styles [15,16,17]; with specific attention to results coming from non-clinical samples, it was found that higher levels of OSAs were associated with higher levels of insecure attachment in Chinese adults with steady partners [18], as well as in a large sample of 14 to 97-year-old Germans [19]. Considering that attachment styles imply specific mental representations about the Self, the others, and their relationships with them, these results stimulate the in-depth exploration of how cybersex users perceive the other people involved in cybersex activities and to which extent they desire a relational connection with them. In this regard, the Internet is a medium that can easily promote the sexual objectification of others (i.e., sexual partners are considered physical objects functional to their own sexual desire, regardless of their whole personality and dignity [20]), and this could be in line with an insecure-avoidant attachment style, in which the desire to avoid relational contact with others is prevalent; nevertheless, some other individuals engage cybersex as an attempt to face loneliness and establish a relational connection with others [21], and this could be in line with an insecure-anxious attachment style, in which seeking contact and appreciation from others is crucial. Finally, another relational aspect concerning cybersex is the perception of infidelity toward steady partners in face-to-face life [22]. In fact, higher levels of OSAs were associated with higher levels of engagement in infidelity-related behaviors on the Internet in both women and men with steady partners from the US [23]. According to Whitty [24], when individuals were presented with a hypothetical scenario of cyber cheating they considered this to be a real form of betrayal that could have a serious impact on offline relationships. Nevertheless, Milheam [25] found that married users of sexual chat rooms were convinced that their online sexual behavior was innocent and harmless. As far as we know, no existing tools are specifically devoted to assessing cybersex users’ beliefs on the betrayal nature of cybersex.

To resume, in the present study, we attempted to fill an existing gap in the literature by developing a new questionnaire to assess beliefs and behaviors concerning emotional and relational aspects in cybersex activities (i.e., the ERACA), as well as by presenting an initial validation in terms of factor structure and internal reliability. Moreover, we presented the associations between the factors of the ERACA and dimensional measures of attachment styles (i.e., predicting that individual differences in secure and insecure attachment styles could account for different cybersex experiences in terms of emotional and relational valence) and online sexual behaviors, including the problematic ones. Finally, we also considered the possible effects of gender, age, and relationship status (i.e., having vs. not having a romantic relationship) in the mean scores of the ERACA factors, as well as in the association between the ERACA factors and the other study variables. These potential moderators were selected because of previous research. Considering that (a) there is evidence suggesting that males and females differ in cybersex attitudes and behaviors (e.g., females were more interested in looking for interactions, while males were more interested in visual-oriented activities [21]); (b) there are age-related differences in the devices used in OSAs (as the age increases, the use of the personal computer increases and the use of mobile devices decreases [1]) that could account for different kinds of cybersex activities; and (c) relationship status impacts on time and consumption of cybersex (e.g., cybersex activity is higher for single users [26]); thus, there are arguments to investigate the moderating role of gender, age, and relationship status.

## 2. Materials and Methods

### 2.1. Participants and Procedure

The present research was realized within a convenience sample of adults contacted in North and Central Italy to take part in research on human sexuality and the Internet. Participants were contacted by trained assistants in public spaces (libraries and cafés) or through social networks and messaging platforms (e.g., different Facebook and WhatsApp groups made up of adults concerning various topics not necessarily connected to cybersex or sexual activities). No economic incentives were given. A total of 488 people were interested in participating and thus were administered the questionnaires. Those contacted in public spaces were first provided with the form containing informed consent, privacy policies, and other research information, and then they were invited to immediately fill out the paper form questionnaires in specially set up stations which allowed for relative quiet and concentration. Those contacted online could directly click on a link to a Google Form containing the informed consent with privacy policies and other research information; after completing the consent form, participants were redirected to a new anonymous page (i.e, not collecting email addresses or other contact data) with the research questionnaires, which was organized in the following way: each questionnaire was presented in a different section so that respondents had to actively move to the next section by clicking on a button; each new section opened with the specific instructions for compiling the questionnaire, and this method is useful to anchor participants’ attention to the specific task and to prevent random answers. Moreover, we paid attention to checking for the carelessness of all respondents by examining the variance of each participant’s responses in each questionnaire. A total of 242 (49.59%) said they had never practiced cybersex activities either alone or with another person, or preferred not to answer specific questions about cybersex. Consequently, the final participants were 246 adults (105 females, 42.68%) from 18 to 65 years of age (mean age = 31.89 years, SD = 10.03 years). A total of 112 (45.53% of the present sample) declared that they have a university degree; 189 (76.83%) reported a romantic relationship.

### 2.2. Measures

#### 2.2.1. Emotional and Relational Aspects in Cybersex Activities—ERACA

The items of the first domain of the ERACA, concerning pleasant and unpleasant emotional activation related to cybersex, were inspired by existing tools (the Internet Sex Screening Test—ISST [27], the Cyber Pornography Addiction Test—CYPAT [28], the Cyber Pornography Use Inventory 9-CPUI-9 [29], and the Problematic Pornography Use Scale-PPUS [16]. In developing the items of this domain, we paid attention to evaluating the emotional activations experienced before (e.g., “*Before having cybersex, I feel excited*”) during (e.g., “*During cybersex, the unpleasant emotions seem to vanish*”), and after (e.g., “*After having had cybersex, I feel depressed and empty*”) cybersex activities, as well as to evaluate whether cybersex allows them to deal with unpleasant moods (e.g., “*I use cybersex as a distraction or an exit strategy from stressful situations*”) and to experience pleasant ones (e.g., “*Cybersex seems to bring excitement into my life*”).

The items of the second domain, concerning how cybersex users relationally perceive the other people involved, were inspired by objectification theory [30], as well as by interview extracts [25] indicating that while some users stated that cybersex was an impersonal activity, some other users declared a feeling of emotional connection. Specifically, in developing the items of this domain we paid attention to evaluating both cybersex users’ general beliefs (e.g., “*During cybersex, it is not necessary to establish a mental or emotional connection with the other person*”) and actual experience (e.g., “*During cybersex, I search for a mental or emotional connection with the other person*”). We note that we paid attention to developing items in such a way that they can be valid either when cybersex is practiced as a solitary activity or it is practiced with others.

Finally, items of the third domain pertained to beliefs concerning the extent to which cybersex is a form of betrayal and infidelity (e.g., “*Cybersex is a form of infidelity and, therefore, it is a threat to the couple*”). They were once again inspired by interview extracts [25].

As a result, a total of 30 items were developed (see Table 1). Each item was posed on a 6-point Likert scale (1 = totally disagree; 2 = strongly disagree; 3 = slightly disagree; 4 = slightly agree; 5 = strongly agree; 6 = totally agree).

#### 2.2.2. Attachment Style Questionnaire—ASQ

The ASQ [31,32] is a 40-item self-report questionnaire made up of five dimensions concerning mental representations about the Self, the others, and the relationships with them. The confidence subscale (8 items, e.g., “*I feel confident that other people will be there for me when I need them*”; Cronbach’s alpha in the present sample = 0.71) reflects the extent to which an individual feels secure in the Self and in relationships; discomfort with closeness (10 items, e.g., “*I worry about people getting too close*”; Cronbach’s alpha in the present sample = 0.73) refers to an insecure-avoidant style characterized by distrust of relationships, linked with the perceived inability to develop closeness with others; relationships as secondary (7 items, e.g., “*Doing your best is more important than getting on with others*”; Cronbach’s alpha in the present sample = 0.78) refers to an insecure-avoidant style described by a hyper-accentuated self-sufficient Self that dismisses the need to get close to others; need for approval (7 items, e.g., “*It’s important to me that others like me*”; Cronbach’s alpha in the present sample = 0.75) is a form of insecure-anxious style in which self-devaluation and the perception of not being worthy of esteem and love activate an exacerbated need for acceptance and confirmation from others; and preoccupation with relationships (8 items, e.g., “*I worry a lot about my relationships*”; Cronbach’s alpha in the present sample = 0.77) represents another facet of insecure-anxious style characterized by fear of abandonment due to the perception of others as unresponsive or inconsistent. Each item was rated on a 6-point Likert scale ranging from 1 (totally disagree) to 6 (totally agree).

#### 2.2.3. Internet Sex Screening Test—ISST

The ISST is a 20-item self-report questionnaire [27] focused on different kinds of online sexual behaviors, including problematic ones. The subscale online sexual compulsivity (6 items, e.g., “*I believe I am an Internet sex addict*”; Cronbach’s alpha in the present sample = 0.80) measures online sexual problems related to the compulsive need to access the Internet for sexual purposes; online sexual behavior—social (5 items, e.g., “*I have participated in sexually related chats*”; Cronbach’s alpha in the present sample = 0.75) concerns the disposition to engage in interpersonal interaction with others during online sexual behaviors; online sexual behavior—isolated (4 items, e.g., “*I have searched for sexual material through an Internet search tool*”; Cronbach’s alpha in the present sample = 0.76) reflects the disposition to engage in solitary online sexual behaviors; online sexual spending (3 items, e.g., “*I have purchased sexual products online*”; Cronbach’s alpha in the present sample = 0.73) assesses the tendency to purchase sexual material; Interest in online sexual behavior (2 items, e.g., “*I have some sexual sites bookmarked*”; Cronbach’s alpha in the present sample = 0.70) assesses the disposition to using computers for sexual aims. The items were rated on a 5-point Likert scale ranging from 0 (never) to 4 (always).

### 2.3. Data Analyses

To investigate the factor structure of the ERACA, we applied an exploratory factor approach to the 30 observed items. First, we explored the form of the distribution of each item using the indices of skewness and kurtosis; scores in the range [−2.00; +2.00] indicate a normal distribution [33]. After that, we verified whether there was a significant number of factors using the Kaiser–Meyer–Olkin’s sampling adequacy criteria (i.e., KMO; values lower than 0.50 are unacceptable [34]) and we tested the hypothesis that correlations between variables were greater than expected by chance adopting Bartlett’s sphericity test (i.e., the related p-value must be significant [35]). To determine the number of factors to extract, we combined several approaches as per the recommendations by [36]: the theory-driven approach (i.e., suggesting the extraction of a number of factors consistent with the theory that has driven the development of the questionnaire), the examination of the scree diagram (i.e., observing how many points are above the point of inflexion in the diagram of the eigenvalues, which represents how much of the variance of the observed variables is explained by each factor), the Kaiser criterion (i.e., suggesting to retain all factors that have an eigenvalue higher than 1.00), and the parallel analysis (i.e., indicating to compare eigenvalues from the EFA with eigenvalues from randomly generated uncorrelated data, and to retain factors with eigenvalues that are greater than the eigenvalues from random data). To avoid distortions due to data distributions, the EFA adopted the principal axis method [37]; a Promax rotation allowed for correlations between latent factors. As for retention criteria, only items with a primary factor loading greater than |0.40| and without cross-loadings greater than 0.32 were retained [37]; moreover, we checked for item redundancy (i.e., items are redundant when they do not add new information) considering inter-item correlations (i.e., the correlations between one item and all other items in the same factor): values above 0.70 may indicate redundancy and the need to carefully consider their simultaneous presence within that dimension [38,39]. Finally, the whole internal consistency of each factor was calculated using Cronbach’s Alpha (although there is not a general consensus, most empirical studies have indicated 0.60 or 0.70 as a minimum standard of reliability [38]).

Considering the ERACA factors that emerged, t-tests for gender (i.e., males vs. females), age (i.e., <mean age vs. >=mean age), and romantic relationship status differences (i.e., presence vs. absence of a current relationship) were inspected. Descriptive statistics of all study variables were calculated (i.e., mean, standard deviation, skewness, and kurtosis), along with zero-order correlations (i.e., Pearson’s *r*). To evaluate the unique contribution of attachment styles and online sexual addition on ERACA factors that emerged, a series of hierarchical regression analyses were performed. Specifically, considering one ERACA factor at a time as a dependent variable, we inserted the other ERACA factors (i.e., to account for their shared variance) and the ASQ dimensions in Step 1. With the aim of investigating the potential interactive role of gender (i.e., males vs. females), age (i.e., <mean age vs. >=mean age), and romantic relationship status differences (i.e., presence vs. absence of a current relationship), we added each potential moderator in 3 different Steps 2, and the 2-way interaction terms between ASQ dimensions and the potential moderators in 3 different Steps 3. The regression approach was re-performed, replacing the ASQ dimensions with the ISST dimensions. Since no significant interaction terms emerged, Steps 2 and 3 were not reported. To highlight and discuss the findings most likely to be meaningful and replicable, significant results in *t*-tests were emphasized only if they presented a practically or clinically significant effect size (i.e., Cohen’s *d* ≥ 0.50; Wolf, 1986); for the same reason, only associations of at least modest effect size (*r* or β ≥ 0.20) were emphasized in the text [40].

## 3. Results

Considering the original 30 items of the ERACA, skewness scores ranged between −0.62 and +1.98, and kurtosis scores ranged between −1.42 and +3.56 (The kurtosis value = 3.56 pertains to item 23. Considering that (a) the skewness value of item 23 was within the range [−2.00; +2.00] (i.e., 1.98), (b) it was the only item with a kurtosis value out of the range [−2.00; +2.00], and (c) the principal axis method used in performing EFA to prevent distortion due to data distribution, we decided neither to delete item 23 from EFA analysis, nor to apply a log transformation to normalize its distribution.) The KMO index was meritorious (0.89). The result of Bartlett’s sphericity test was χ^2^ = 3820.03, df = 435; *p* < 0.001. The inspection of the scree diagram indicated the extraction of five factors (i.e., corresponding to those with eigenvalues greater than 1.00), while the parallel analysis indicated that four factors presented eigenvalues greater than those from the random data. Considering that a 4-factor solution resulted in one factor with only two items, and according to the 3-factor theory-driven model, we opted for a parsimonious approach and we extracted three factors (51.25% of explained variance). Results of the CFA are reported in Table 1. Factor 1 comprised 7 items that refer to cybersex as a means to get excitement and satisfaction without the need or the desire to make emotional or relational connections with other people involved in cybersex activities, and it was labeled “gratification of the Self through objectification of other people”; it accounted for 30.10% of variance and its factor loadings ranged from |0.54| to |0.80|; Cronbach’s Alpha was 0.84, and values of inter-item correlations did not exceed |0.56|. Factor 2 was made up of 8 items on the use of cybersex to deal with negative emotional states and search for positive ones in order to gratify one’s own Self through the relational aspects of the cybersex activities. It was labeled “gratification of the Self through relational aspects”, it accounted for 13.80% of the variance and its factor loadings ranged from |0.53| to |0.68|; Cronbach’s Alpha was 0.85, and values of inter-item correlations did not exceed |0.62|. Lastly, factor 3 comprised 4 items concerning beliefs about cybersex as a potential betrayal for romantic relationships (i.e., when only one member of the couple is involved in cybersex), and feelings of shame and guilt because of cybersex activities. It was labeled “betrayal and infidelity” and it accounted for 7.36% of the variance; its factor loadings ranged from |0.47| to |0.74|; Cronbach’s Alpha was 0.70, and values of inter-item correlations did not exceed |0.65|.

No gender, age, or romantic relationship status differences emerged. Descriptive statistics and zero-order correlations were reported in Table 2. We note that higher levels in the factor gratification of the Self through objectification of other people were positively associated with the factor gratification of the Self through relational aspects (*r* = 0.49, *p* < 0.001) and negatively associated with the factor of betrayal and infidelity (*r* = −0.22, *p* < 0.001).

The results of regression analyses are reported in Table 3. Note that the regression approach herein adopted allowed to control for the shared variance between ERACA factors (i.e., consider the notable positive associations between the first and the second ERACA factor). Higher levels of discomfort for closeness (β = 0.20, *p* < 0.01) and online sex behavior—isolated (β = 0.37, *p* < 0.001) were related to higher levels of using cybersex as a means to reach gratification of the Self through objectification of other people. Moreover, lower levels of discomfort with closeness (β = −0.21, *p* < 0.01) and higher levels of considering relationships as secondary (β = 0.26, *p* < 0.001), needing approval (β = 0.22, *p* < 0.01), and online sex—social (β = 0.31, *p* < 0.001) were associated to higher levels of using cybersex as a mean to reach gratification of the Self through relational aspects. Finally, lower levels of online sex behavior isolated (β = −0.24, *p* < 0.01), and higher levels of online sex compulsivity (β = 0.28, *p* < 0.001), and online sex behavior—social (β = 0.22, *p* < 0.01) were related to higher levels in considering cybersex a betrayal and infidelity activity.

## 4. Discussion

The present research was realized to explore in-depth individual differences in emotional and relational aspects related to cybersex activities by presenting the development and the initial validation of a new questionnaire, i.e., the Emotional and Relational Aspects in Cybersex Activities—ERACA. After the development of a pool of items inspired by extant tools or research reporting qualitative interviews, an EFA approach conducted within a community sample of adults indicated the presence of three factors, each showing satisfying internal reliability. Subsequent associations with measures of both attachment styles and involvement in online sexual behaviors were explored.

The first factor emerging, named “gratification of the Self through objectification of other people”, refers to experiencing cybersex as a means to get excitement and satisfaction without the need or the desire to make emotional or relational connections with others, who are objectified. Significant differences in this factor were accounted for by the facet of insecure-avoidant attachment style defined by distrust of relationships linked with a perceived inability to develop closeness with others, as well as by the disposition to engage in solitary cybersex activities behavior. While the positive association with a measure of insecure attachment was not surprising and in line with previous research [18,19], our results may suggest a particular emotional and relational profile associated with cybersex users high in this factor. Specifically, although the cross-sectional nature of this study does not permit causal conclusions, we could assume that cybersex users high in this factor desire to engage in solitary sexual activities that they perceive as “secure”, avoiding getting relationally involved; in line with this, objectification of the other people involved in cybersex activities could be an avoidant strategy that serves to alleviate insecurity in relationships. This could be also in line with failure in past relational experiences (i.e., see Cooper et al.’s classification of profiles of OSA users [2]), and it would be interesting to directly assess this aspect in future research.

The second factor that emerged, named “gratification of the Self through relational aspects”, refers to experiencing cybersex to deal with negative emotional states and search for positive ones, in which its relational aspects are functional to gratify one’s own Self. Differently from the first factor, its variance was accounted for by the disposition to engage in interpersonal interaction with others during cybersex, as well as by an articulated pattern regarding attachment styles. First, both the two facets of the insecure-avoidant style presented significant and unique associations, even if they were opposite in direction: while avoidance related to a perceived inability to develop closeness was negatively associated with this factor, avoidance related to a hyper-accentuated self-sufficient Self that denies the needs of others presented a positive association. Moreover, the facet of insecure-anxious attachment related to an exacerbated need for acceptance and confirmation from others associated with self-devaluation and the perception of not being worthy of esteem and love on the part of others presented a positive association with this second factor. Despite all the precautions advanced above in interpreting cross-sectional results, this evidence may suggest that high levels in getting gratification of the Self through the relational aspects of cybersex could be the results of different pathways: on the one hand, the relational opportunities of cybersex could be recognized and appreciated in a utilitarian and self-centered way (i.e., an avoidant dismissing approach); on the other hand, individuals with high anxiety in social relationships could approach cybersex as an opportunity to expand their relationship-seeking strategies and reaching gratification for a fragile Self that needs continuous confirmation from others (i.e., a preoccupied anxious approach, that could be in line with Cooper’s profiles of users that approach OSAs as a coping strategy to deal with unpleasant or overwhelming emotions [2]).

Finally, the third factor, named “betrayal and infidelity”, accounts for considering cybersex as a potential betrayal for romantic relationships when only one member of the couple is involved in this kind of activity. While no significant associations on the part of attachment styles emerged, there emerged the unique positive contribution of online sexual compulsivity and online sexual behavior—social. In other words, the perception of infidelity related to cybersex could be specifically developed in the context of repeated problematic experiences (i.e., when cybersex interferes with other areas of life, including offline romantic relationships), as well as in the context of seeking interaction with other people during cybersex activities (i.e., when it might be more likely to develop deeper knowledge and/or relationships with others; in this regard, note the negative correlation between this factor and the first factor, concerning the tendency to objectify other people). Moreover, there emerged the unique negative contribution of online sexual behavior - isolated: in line with the construct of behavioral rationalization [25], we could hypothesize that individuals who mainly practice solitary cybersex apply a disengagement process convincing the Self that ethical standards do not apply to themselves in the context of impersonal sexual activities.

Our findings must be read within the context of some limitations. The first limitation pertains to external validity and calls into question the participants of the present study. Specifically, we used a convenience sample that was in part recruited online, and this could have affected the generalizability of the results to the entire population (e.g., those who use the Internet to practice cybersex but do not habitually use social media); moreover, the research was conducted in a homogeneous cultural context (i.e., the Italian one), and thus the generalizability of the results to other cultures should be investigated. Future studies should adopt a sampling technique capable of reaching as many types of cybersex users as possible, representative of the entire adult population, and should involve participants coming from other countries and cultures. Further, we did not collect information about the quantitative and qualitative use of cybersex activities, and future research could investigate its potential moderating effect on the investigated variables. Importantly, the cross-sectional nature of the present study did not allow causal conclusions, and only future longitudinal studies could clarify in-depth the causal patterns about the association between our variables; in this regard, we have to also note that no gender, age, or romantic relationship status moderating effects emerged: on the one hand, this could indicate that the associations we found refer to general processes that are independent of these individual differences; on the other hand, we believe it would be important to further test this point—maybe from a longitudinal perspective and also with reference to the association between the ERACA variables and a wider range of variables that refer to the quality of online and offline relations. For instance, we did not consider participants’ sexual satisfaction or offline sexual relationships, and future research should explore their association with the perception of betrayal and infidelity related to cybersex. In addition, we presented here an EFA approach: it would be important to further investigate the psychometric properties of the ERACA by providing a confirmatory factor analysis approach that would allow in-depth evidence of its factorial validity (e.g., maybe also testing the invariance of its factor structure with reference to gender, age, and romantic relationship status). Lastly, we must note that the emerging factors did not concur with the factors identified in our theoretical model: specifically, we found that there is not a strict division between emotional and relational aspects related to cybersex, since both factors related to the gratification of the Self (i.e., through the objectification of other people or through valorizing relationships) include both emotional and relational aspects.

## 5. Conclusions

Despite these limitations, we believe that the present study has its strength in providing a brief and easy-to-use tool that allows us to assess individual differences in emotional and relational aspects related to cybersex activities. The usefulness of the ERACA is twofold. First, it can be applied for research purposes to better understand the dynamics relating to the phenomenon of cybersex and its associations with the quality of relationships. For instance, the associations we found with the attachment variables indicating that there may be differential pathways that lead to the gratification of the Self through the relational aspects of the cybersex need further investigation to better understand the different valence that cybersex activities assume in people’s lives. Second, from a health-promoting point of view, it can be adopted for screening purposes within the general adult population, especially when research will have provided evidence of associations with other instruments focused on the quality of relationships in (sub)clinical context to better understand to what extent cybersex activities could constitute a risk factor for their well-being.

## Figures and Tables

**Table 1 ijerph-20-07151-t001:** The original 30 items of the ERACA (the original Italian form is enclosed in squared brackets) and the results of the EFA.

Item Content	Theoretical Domain	Factor Loadings (λ) from EFA		
Please, fill out the following section on the basis of your experience about cybersex (i.e., sexual activities carried out through the use of the Internet). The following statements refer to the emotional and relational aspects of cybersex. [Compili questa sezione in base alla sua esperienza di sesso online su internet, ovvero di attività sessuali realizzate attraverso l’uso di internet. Le seguenti affermazioni fanno riferimento alla sfera emotiva e relazionale del sesso online.]		Factor 1: Gratification of the Self through objectification of other people	Factor 2: Gratification of the Self through relational aspects	Factor 3: Betrayal and infidelity
1. I find it satisfactory that cybersex does not result in a relational engagement. [Trovo soddisfacente che il sesso online non si traduca in un impegno relazionale.]	Relational aspects	**0.578**	−0.112	−0.023
2. Before having cybersex, I feel excited. [Prima di fare sesso su internet mi sento eccitato/a.]	Emotional activation	0.398	0.421	−0.016
3. During cybersex, I think of the other person as an object to achieve sexual gratification. [Durante le attività di sesso online penso all’altro come ad un oggetto per raggiungere la gratificazione sessuale.]	Relational aspects	**0.670**	0.140	0.075
4. Having cybersex makes me feel less lonely. [Fare sesso online mi fa sentire meno solo/a.]	Emotional activation	0.096	**0.624**	0.174
5. During cybersex, the unpleasant emotions seem to vanish. [Durante il sesso online le emozioni spiacevoli sembrano svanire.]	Emotional activation	0.217	**0.614**	−0.011
6. In cybersex, people use each other. [Nel sesso online le persone si usano a vicenda.]	Relational aspects	0.753	−0.063	0.337
7. Cybersex is a harmless form of sexual entertainment that does not pose a threat to the couple. [Il sesso su internet è un’innocua forma di intrattenimento sessuale che non rappresenta una minaccia per la coppia.]	Betrayal and infidelity	0.245	0.164	**−0.465**
8. Cybersex makes me feel important and complete. [Il sesso online mi fa sentire importante e completo/a.]	Emotional activation	−0.021	**0.684**	−0.036
9. After having had cybersex, I feel depressed and empty. [Dopo il sesso su internet mi sento depresso/a e vuoto/a.]	Emotional activation	0.330	0.130	0.563
10. During cybersex, the perception of the other person is reduced to parts of her/his body. [Durante il sesso online la percezione dell’altro/a si riduce alle parti del suo corpo.]	Relational aspects	**0.800**	−0.147	0.227
11. Cybersex is a violation of trust between the members of the couple. [Fare sesso online costituisce una violazione della fiducia della coppia.]	Betrayal and infidelity	0.100	−0.007	**0.740**
12. I use cybersex as a form of entertainment. [Uso il sesso su internet come forma di intrattenimento.]	Emotional activation	**0.669**	0.075	0.003
13. Cybersex makes me feel attractive and desired. [Il sesso su internet mi fa sentire attraente e desiderato/a.]	Emotional activation	0.031	**0.650**	0.237
14. During cybersex, I feel pleasant emotions. [Durante il sesso su internet provo emozioni piacevoli.]	Emotional activation	0.340	0.533	−0.142
15. I use cybersex as a distraction or an exit strategy from stressful situations. [Uso il sesso online come distrazione o fuga da situazioni stressanti.]	Emotional activation	0.418	0.366	0.049
16. After having had cybersex, I feel excited. [Dopo il sesso online mi sento eccitato/a.]	Emotional activation	0.140	**0.604**	−0.223
17. During cybersex, the other is a person with her/his own identity, her/his own life story, etc. [Durante il sesso su internet l’altro/a è una persona con una propria identità, una propria storia di vita, ecc.]	Relational aspects	−0.144	**0.534**	0.222
18. During cybersex, it is not necessary to establish a mental or emotional connection with the other person. [Durante il sesso online non è necessario stabilire un contatto mentale o emotivo con l’altra persona.]	Relational aspects	**0.730**	−0.277	−0.001
19. I find it frustrating that cybersex does not result in a relational engagement. [Trovo frustrante che il sesso online non si traduca in un impegno a livello relazionale.]	Relational aspects	−0.160	0.381	0.368
20. In cybersex, the other person is a product of my desires, an illusion. [Nel sesso su internet l’altro/a è un prodotto dei miei desideri, un’illusione.]	Relational aspects	**0.757**	−0.128	0.078
21. During cybersex, I feel arousal. [Durante il sesso su internet provo eccitazione.]	Emotional activation	**0.543**	0.292	−0.142
22. During cybersex, I idealize and fantasize about the other person. [Durante gli atti sessuali online idealizzo e faccio fantasie sull’altro/a.]	Relational aspects	0.372	0.406	0.024
23. Before having cybersex, I experience unpleasant emotions such as fear, anxiety, and anger. [Prima di fare sesso su internet provo emozioni spiacevoli come ad esempio paura, ansia, rabbia.]	Emotional activation	0.011	0.336	0.382
24. Cybersex seems to bring excitement into my life. [Il sesso online sembra introdurre eccitazione nella mia vita.]	Emotional activation	0.139	**0.635**	−0.072
25. During cybersex, I search for a mental or emotional connection with the other person. [Durante il sesso su internet ricerco un contatto mentale o emotivo con l’altro/a.]	Relational aspects	−0.370	0.815	0.273
26. After having had cybersex, I feel gratification and I feel satisfied. [Dopo il sesso online provo gratificazione e mi sento soddisfatto/a.]	Emotional activation	0.118	**0.569**	−0.264
27. Cybersex does not imply physical contact with the other person, so it does not constitute a form of betrayal. [Il sesso su internet non prevede il contatto fisico con l’altra persona, pertanto non costituisce una forma di tradimento.]	Betrayal and infidelity	0.341	0.126	−0.376
28. During cybersex, I perceive the other as a person, with her/his own feelings, thoughts, etc. [Durante il sesso online percepisco l’altro come persona, con propri sentimenti, pensieri, ecc.]	Relational aspects	−0.397	0.769	0.108
29. After having had cybersex, I feel ashamed and guilty. [Dopo il sesso online provo vergogna e mi sento in colpa.]	Emotional activation	0.243	0.179	**0.605**
30. Cybersex is a form of infidelity and, therefore, it is a threat to the couple. [Il sesso su internet rappresenta una forma di infedeltà e quindi una minaccia per la coppia.]	Betrayal and infidelity	0.016	−0.005	**0.651**

Note: factor loadings in bold indicate to which factor each item was attributed.

**Table 2 ijerph-20-07151-t002:** Descriptive statistics and zero-order correlations (Pearson’s *r*).

	M (SD)	Skew.	Kurt.	1	2	3	4	5	6	7	8	9	10	11	12	13	14
ERACA—Gratification of the Self through objectification of other people	3.42 (1.29)	−0.25	−0.78	-													
ERACA—Gratification of the Self through relational aspects	2.55 (1.10)	0.35	−0.67	0.49 ***	-												
ERACA—Betrayal and Infidelity	2.87 (1.25)	0.30	−0.79	−0.22 ***	−0.04	-											
ASQ—Confidence	3.40 (0.71)	−0.43	0.44	0.05	−0.06	−0.09	-										
ASQ—Discomfort with Closeness	3.69 (0.75)	0.07	0.04	0.12	0.01	0.06	−0.43 ***	-									
ASQ—Relationships as Secondary	2.40 (0.84)	0.72	0.38	0.22 ***	0.36 ***	−0.02	−0.23 ***	0.26 ***	-								
ASQ—Need for Approval	3.00 (0.91)	0.20	−0.40	0.01	0.24 ***	0.13 *	−0.33 ***	0.31 ***	0.16 **	-							
ASQ—Preoccupation with Relationships	3.50 (0.89)	−0.11	0.03	−0.01	0.11	0.15 *	−0.25 ***	0.35 ***	0.04	0.54 ***	-						
ISST—Online sexual compulsivity	0.47 (0.66)	1.94	3.71	0.26 ***	0.42 ***	0.14 *	−0.25 ***	0.09	0.40 ***	0.22 ***	0.18 **	-					
Online sexual behavior—social	0.66 (0.75)	1.29	1.47	0.20 **	0.49 ***	0.13 *	−0.10	−0.01	0.34 ***	0.10	0.11	0.55 ***	-				
ISST—Online sexual behavior—isolated	1.98 (1.01)	−0.05	−0.67	0.46 ***	0.32 ***	−0.21 ***	−0.07	0.06	0.22 ***	0.07	0.08	0.39 ***	0.40 ***	-			
ISST—Online sexual spending	0.31 (0.62)	2.59	7.68	0.24 ***	0.40 ***	−0.03	−0.09	−0.08	0.39 ***	0.11	−0.01	0.54 ***	0.50 ***	0.28 ***	-		
ISST—Interest in online sexual behavior	0.68 (1.01)	1.66	1.99	0.27 ***	0.41 ***	−0.10	−0.03	0.004	0.25 ***	0.06	0.02	0.47 ***	0.46 ***	0.52 ***	0.52 ***	-	
SASTA—Sex addiction	0.70 (0.71)	1.50	2.29	0.23 ***	0.31 ***	0.21 ***	−0.31 ***	0.17 **	0.34 ***	0.25 ***	−0.15 *	0.67 ***	0.47 ***	0.37 ***	0.37 ***	0.37 ***	-

Note: * *p* < 0.05, ** *p* < 0.01, *** *p* < 0.001.

**Table 3 ijerph-20-07151-t003:** Regression analyses (the ERACA factors are the dependent variables).

	ERACA—Gratification of the Self through Objectification of Other People	ERACA—Gratification of the Self through Relational Aspects	ERACA—Betrayal and Infidelity
Predictors: ASQ variables	F(7245) = 16.315, *p* < 0.001; R^2^ = 0.30	F(7245) = 16.138, *p* < 0.001; R^2^ = 0.36	F(7245) = 2.874, *p* < 0.01; R^2^ = 0.05
ERACA—Gratification of the Self through objectification of other people	-	0.47 ***	−0.25 ***
ERACA—Gratification of the Self through relational aspects	0.51 ***	-	0.06
ERACA—Betrayal and Infidelity	−0.18 **	0.04	-
ASQ—Confidence	0.11	−0.02	−0.02
ASQ—Discomfort with Closeness	0.20 **	−0.21 **	0.03
ASQ—Relationships as Secondary	0.03	0.26 ***	−0.01
ASQ—Need for Approval	−0.11	0.22 **	0.05
ASQ—Preoccupation with Relationships	−0.03	0.05	0.10
Predictors: ISST variables	F(7245) = 20.819, *p* < 0.001; R^2^ = 0.36	F(7245) = 26.324, *p* < 0.001; R^2^ = 0.42	F(7245) = 6.881, *p* < 0.001; R^2^ = 0.14
ERACA—Cybersex as a means to reach satisfaction through objectification of other people	-	0.40 ***	−0.16 *
ERACA—Gratification of the Self through relational aspects	0.44 ***	-	−0.03
ERACA—Betrayal and Infidelity	−0.12 *	−0.02	-
ISST—Online sexual compulsivity	0.04	0.11	0.28 ***
ISST—Online sexual behavior—social	−0.15 *	0.31 ***	0.22 **
ISST—Online sexual behavior—isolated	0.37 ***	−0.12	−0.24 **
ISST—Online sexual spending	0.06	0.05	−0.13
ISST—Interest in online sexual behavior	−0.10	0.15 *	−0.09

Notes: * *p* < 0.05, ** *p* < 0.01, *** *p* < 0.001.

## Data Availability

There are no unpublished data available. The corresponding author can be contacted on this matter.

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
