# Peer review of "Cybersex and Attachment Styles: Proposal of the Emotional and Relational Aspects in Cybersex Activities (ERACA) Questionnaire"

_ijerph, 2023, doi:10.3390/ijerph20247151_

Round 1
Reviewer 1 Report
Comments and Suggestions for Authors
This is an enlightening study, however, there are limitations as follows:
1.In the introductory section, the importance and significance of why this study was undertaken needs to be reinforced.
2. Literature research in this area is not sufficient and some important relevant literature is not covered, therefore, further strengthening of literature research and analysis is needed.
3. In section 2.1. Participants and Procedure, it is necessary to focus on every detail of the questionnaire, especially in relation to sensitive issues, and how to ensure that the respondent's answers are truthful and credible.
4. In the discussion section, the issue of external validity of this study needs to be introduced.
Comments on the Quality of English Languageno
Reviewer 2 Report
Comments and Suggestions for Authors
ERACA is a concise and easy-to-use instrument which has sufficient internal consistency and construct validity.
However, it should be noted that:
1. The rationale for selecting the three issues examined in the questionnaire lacks conviction.
2. The questionnaire is based on a 5-point scale including the following options: 1 = totally disagree; 2 = strongly disagree; 3 = slightly disagree; 4 = slightly agree; 5 = strongly agree; 6 = totally agree. Such a fractional scale, on the one hand, is an advantage of the measure since it allows the researcher to obtain differentiated evaluation of the aspects under measurement. On the other hand, excessive detailing can be inconsistent with respondents' ability to make such fine distinctions in the level of agreement, which will result in overestimating the consistency of responses due to attitudinal factors.
3. The items in the questionnaire have a forward and backward key. But the correlation of the reverse items with the total score on the questionnaire was not analyzed. This means it is not entirely clear whether there are shifts in respondents' evaluations under the influence of the consent attitudes.
4. Unfortunately, there is no indication of how the criterion validity of the questionnaire has been tested.
Reviewer 3 Report
Comments and Suggestions for Authors
factor analysis to extract three factors from the Emotional and Relational Aspects in Cybersex Activities (ERACA) questionnaire. They also collected data using the Attachment Style Questionnaire (ASQ) and the Internet Sex Screening Test (ISST).
The researchers presented data supporting the validation of the three structure model of the ERACA and they regressed each ERACA factor and presumed predictor variables including the ASQ and ISST.
Overall, the study is interesting and contemporary. The authors made a strong case for the study by describing the emergence of cybersex as an important emerging subject of study. The data analyses they presented support their case and makes an important contribution.
The main weaknesses of the study are summarized by two things.
First, the authors could describe in more detail how participants were recruited for the study. They stated that trained professionals contacted 488 participants who indicated an interest in participating in a study from Facebook and WhatsApp and 246 completed the questionnaire. However, readers would like to know more about how initial recruitment took place. Was is Facebook or WhatsApp advertising? Or, was it some other way. Moreover, were the questionnaires mailed to participants or did they take an online survey. There are more details that could have been provided in the methods section.
Second, the authors stated that no gender or age differences were observed on Page 5 and excluded those from the analyses. It is likely that these demographic variables are important and previous literature supports the inclusion of these variables. If they were not significant, the researchers should consider describing them and including an explanation in the discussion. In the same way, very little is known about the offline sexual relationships with the participants. There is simply a binary variable indicating whether or not they had a significant relationship. Although betrayal and infidelity as an important factor in the ERACA, the readers don’t have much insight on the sexual or romantic relationships of the participants.
Round 2
Reviewer 1 Report
Comments and Suggestions for Authors
Overall, there is an improvement in quality over previous manuscript.
Comments on the Quality of English LanguageMinor editing of English language required